# Microstructure and Properties of Ti(C,N)-Based Cermets with Al_x_CoCrFeNiTi Binder

**DOI:** 10.3390/ma16072894

**Published:** 2023-04-05

**Authors:** Meiling Liu, Zhen Sun, Peng Liu, Wanxiu Hai, Yuhong Chen

**Affiliations:** 1College of Materials Science and Engineering, North Minzu University, Yinchuan 750021, China; 2Key Laboratory of Powder Material and Advanced Ceramics, North Minzu University, Yinchuan 750021, China; 3International Scientific and Technological Cooperation Base of Industrial Waste Recycling and Advanced Materials, North Minzu University, Yinchuan 750021, China

**Keywords:** Ti(C,N) cermet, microstructure, mechanical properties, oxidation resistance

## Abstract

Al_x_CoCrFeNiTi (x = 0.1, 0.3, 0.6, 1) powders were prepared via mechanical alloying and were used as binders for SPS-produced Ti(C,N)-based cermets. The effects of AlxCoCrFeNiTi binder on phase composition, morphology, room-temperature mechanical properties, and oxidation resistance of cermets were studied. The research showed that cermets with Al_x_CoCrFeNiTi binders exhibited a more homogeneous core–rim structure than cermets with cobalt binders. The Vickers hardness and fracture toughness of cermets with Al_x_CoCrFeNiTi binders increased with the aluminum molar ratio due to the grain refinement and solid solution strengthening effect of carbonitrides. After static oxidation at 1000 °C, the mass gain of the cermets with Al_x_CoCrFeNiTi binders changed according to a quasi-parabolic law, and the lowest mass gain was obtained in the cermet with Al_0.6_CoCrFeNiTi binder. The oxidation kinetics curve of the benchmark cermet with cobalt followed a linear law. The oxidation product of Ti(C,N)-based cermet with cobalt was rich in TiO_2_, and the Ti(C,N)-based cermets with Al_x_CoCrFeNiTi binders were transformed into complex oxides, such as NiMoO_4_, NiWO_4_, FeMoO_4_, Fe_3_Ti_3_O_9_, and Ni_3_TiO_7_. The oxide layer on the cermet with Al_0.6_CoCrFeNiTi appeared to be dense and protective, which inhibited the diffusion of oxygen into the cermet and improved the oxidation resistance of the final product.

## 1. Introduction

Thanks to outstanding red hardness, thermal stability and wear resistance, Ti(C,N)-based cermets have been utilized as cutting tool materials to substitute cemented carbides [1,2,3]. As a basic component of cermets, metal binder has been considered as the main factor influencing density, strength, and toughness [4,5]. To withstand severe working conditions with elevated temperatures up to 600–1000 °C, the cermet tools must have sufficiently high thermomechanical properties, tribological properties, and thermal oxidation resistance. However, the traditional binders, such as Co and Ni, were found to limit the further application of cermets [6,7,8].

Other alternative binders have been investigated in last decades [9,10,11], among which much attention has been paid to the high-entropy alloys (HEAs) because of their impressive hardness and distinguishing thermal stability [12,13]. High-entropy alloys (HEAs) are mainly solid solutions, containing five or more principal elements in equal or near-equal atomic percentages (5–35 at.%) [12]. Because of four core effects, many attempts have been made to apply HEAs to enhance the mechanical performance, wear resistance and oxidation resistance of Ti(C,N) cermets. Zhu et al. [14] prepared a Ti(C,N)-AlCoCrFeNi cermet and showed co-enhancement of hardness and toughness. Wang et al. [15] fabricated a Ti(C,N)-CoCrFeNiCu cermet with increased Vickers hardness, fracture toughness, transverse rupture strength, and wear performance relative to those of Ti(C,N) with Ni binder. Zhu et al. [16] reported the Ti(C,N)-AlCoCrFeNi cermet, the high-temperature (1000 °C) oxidation performance of which exceeded that of Ti(C,N)-Ni/Co cermet in terms of both mass gain and reaction rate. Ji et al. [17] synthesized the TiB_2_-5wt.% AlCoCrFeNiTi ceramics with highly improved mechanical properties, especially flexural strength. Fu et al. [18] compared the mechanical properties of ultrafine TiB_2_-AlCoCrFeNiTi cerments with that of TiB_2_-Ni binders. Li et al. [19] found that the wear rate of Ti(C,N)-TiB_2_ cermet with FeCoCrNiAl binder at 600 °C and 800 °C was 39.25% and 46.7% lower than that of the Ni-Co binder cermets, respectively. Guo et al. [20] produced a CoCrFeNi-based Ti(C,N) cermet via NbC addition with a considerably enhanced mechanical performance and a high-temperature wear rate. Fang et al. [21] prepared a Ti(C,N) cermet with Al_0.3_CoCrFeNi binder. This exhibited better high-temperature mechanical properties, oxidation resistance and wear resistance than those of Ni-Co binder. Until now, the research into Ti(C,N) cermets made with HEA binders has focused on mechanical and tribological performance, whereas oxidation resistance has been little investigated. Moreover, the impact of Al_x_CoCrFeNiTi binder on the oxidation resistance of Ti(C,N) cermets is still poorly understood.

In this work, Al_x_CoCrFeNiTi (x = 0.1, 0.3, 0.6, 1) HEA powders were fabricated as binders to be added into Ti(C_0.7_, N_0.3_)-WC-Mo_2_C-HEAs cermets. The influence of HEA addition to the microstructure, mechanical properties, and high-temperature oxidation resistance of cermets was discussed as well.

## 2. Materials and Methods

### 2.1. Fabrication of HEAs

The metal powder raw materials, including aluminum, nickel, cobalt, chromium, iron and titanium (all with the average size of 30 μm) were purchased from Hunan Three Six Nine Metallurgical Technology Limited Co., Ltd. (Changsha, China). First, the metal powders were weighed according to a molar ratio of Al:Co:Cr:Fe:Ni:Ti as (0.1, 0.3, 0.6, 1):1:1:1:1:1 and then mechanically alloyed with a planetary ball mill to obtain the Al_x_CoCrFeNiTi HEAs powders. Each of six kinds of raw powders was placed into a steel vial and milled with a powder-to-ball ratio of 10:1 for 12 h at 450 rpm under Ar atmosphere.

### 2.2. Preparation of Cermets

The commercial ceramic raw materials (Sinopharm Group, Beijing, China) included Ti(C_0.7_, N_0.3_) (1–2 μm), WC (2 μm), and Mo_2_C (2 μm). The basic compositions (wt.%) of Ti(C,N)-based cermets were made according to Ti(C_0.7_, N_0.3_)-20%WC-10%Mo_2_C-10% HEAs. The weighed raw materials were mixed via ball milling with a powder-to-ball ratio of 10:1 for 12 h at 200 rpm under an Ar atmosphere. Ti(C,N)-based cermets with HEA molar ratios of 0.1, 0.3, 0.6 and 1 were produced via spark plasma sintering and referred to as TNAl1, TNAl2, TNAl3 and TNAl4, respectively. Meanwhile, the benchmark cermet with Co binder, named TNCo, was prepared through the same process. The sintered samples were ground and polished for the following tests and characterizations.

### 2.3. Test and Characterization

The polished cermets were placed in a Muffle furnace and heated at 1000 for 4 h to carry on the oxidation test in the air. The phase structures of HEA powders after mechanical alloying and sintered cermets were examined via X-ray diffraction with Cu-Kα(XRD, Shimadzu-6000, Japan). A scanning electron microscope (SEM, Zeiss SIGMA 500, Germany) was employed to observe the microstructure and morphology in the device’s backscatter mode, and element contents of cermets were assessed using an energy-dispersive spectrometer (EDS). The average Ti(C,N) grain size in the cermets was measured from the SEM micrographs using the linear interception method. The cermets were cut perpendicularly after high-temperature oxidation to observe their cross-sectional oxidation morphology. Transmission electron microscopy (TEM, Tecnai G2 F20, FEI, USA) was employed to further confirm the microstructure. The Vickers hardness was measured according to GB/T 37900-2019 standard on a Wolpert-432SVD hardness tester under a load of 30 kg applied for 15 s. The fracture toughness (*K*_IC_) was then calculated as follows [22]:(1)$$\begin{array}{l} K_{\mathrm{IC}}=0.15\sqrt{\frac{H_{V30}}{\sum_{i=1}^{4}L_{i}}}\\ \overset{-}{1}1\overset{-}{1} \end{array}$$
where *H*_V30_ is the Vickers hardness (Kgf/mm^2^), and *L_i_* is the crack length (mm) in the 500× optical microscope.

## 3. Results

### 3.1. Phase Composition

The XRD patterns of the Al_x_CoCrFeNiTi HEA powders with molar ratios of Al 0.1, 0.3, 0.6 and 1 are illustrated in Figure 1a.

For the HEAs powder with the Al content of 0.1, the main phase was an FCC alloy phase with a small amount of titanium. With the increase in Al amount to 0.3, besides the above phases, there existed a low quantity of BCC alloy. Once the Al molar ratio increased from 0.6 to 1, the numbers of peaks of BCC alloy and their intensities increased, but the FCC phase was still predominant. The BCC-to-FCC peak area ratios in Al_0.3_CoCrFeNiTi, Al_0.6_CoCrFeNiTi and AlCoCrFeNiTi cermets were 6.3%, 18.5% and 22.6%, respectively. The XRD patterns of sintered Ti(C,N) cermets with HEA binders and benchmark cermet with cobalt binder are depicted in Figure 1b. Compared with other three cermets with HEAs, TNAl1 exhibited the strongest peak associated with the undissolved WC phase, and the undissolved Mo_2_C also existed therein. With the increase in Al content from 0.1 to 0.6, the WC peaks decreased until vanishing, but the peak intensity of the undissolved Mo_2_C increased. Once the Al molar ratio reached 1, none of the WC- and Mo_2_C-related peaks were detected, which meant that WC and Mo_2_C were almost solutionized. With the increase in Al content of HEAs, the solid solution of Mo_2_C got first worse and then better, while that of WC was continuously getting better. This indicated that the formation of WC and Mo_2_C solid solutions was promoted by the generation and enlargement of HEAs’ BCC phase. The reason was probably that lattice distortion become more severe with the increase in the Al content [23]. Along with a sluggish diffusion effect, it accelerated the dissolution of carbides and precipitation of carbonitrides. The bimodal peak at a little lower angle next to the Ti(C,N) main peak was ascribed to a (Ti, W, Mo)(C, N) solid solution. The (Ti, W, Mo)(C, N)-to-Ti(C,N) peak area ratios of TNAl1, TNAl2 and TNAl3 were 9.8%, 12.3% and 16.6%, respectively. The increasing amounts of solutionization products were in line with the decrease in carbides in cermets with HEA binders. Moreover, the peak intensity of WC in the benchmark TNCo cermet was much higher than those of the other four cermets with HEAs binders. Thus, it was concluded that the solid solution formation of Ti(C,N) and carbides was significantly promoted by HEAs.

### 3.2. Morphology and Microstructure of Cermets

Figure 2 displays the SEM images of five Ti(C,N) cermets with a typical core–rim structure.

As seen in Figure 2a, there existed the coarse Ti(C,N) hard phase particles (black) with some large particles aggregating in TNAl1 cermet, while the solid solution rims (gray) were in a minority. The irregular paillettes (white) were undissolved WC inclusions, which were confirmed by the EDS analysis of point 2 in Figure 2b. Meanwhile, the composition of rim at point 1 was ascribed to the (Ti,W,Mo)(C,N) solid solution in Figure 2b. According to Figure 2c, the Ti(C,N) particles became finer, and paillettes were hardly found in TNAl2 in the low-magnification images. Given that the numbers of XRD peaks of WC phase and their intensities drastically decreased, the solid solution degree was further increased. As seen in Figure 2d, the Ti(C,N) hard phase particles (black) were still fine, but there also existed flakes (gray). Based on the XRD peaks and EDS analysis of point 3 in Figure 2e, flakes were the mixture of Mo_2_C and (Ti,W,Mo)(C,N) solid solutions. As shown in Figure 2f, compared to the first three cermets, TNAl4 had a typical and homogeneous core–rim structure as TNAl3. According to the XRD picture, WC and Mo_2_C were solutionized sufficiently. In Figure 2g, abundant WC paillettes (white) and the ununiform core–rim structure were observed in the benchmark TNCo cermet. The Ti(C,N) grain sizes of the five cermets measured were 0.889 µm, 0.806 µm, 0.774 µm, 0.720 µm and 0.93 µm, respectively.

The HAADF images and EDS maps of TNAl3 cermet are shown in Figure 3, and the selected area as square area in Figure 3a was subjected to the HRTEM analysis in Figure 4. As seen in Figure 3, the black core phase was surrounded by thin gray rims, and the HEA binder appeared in bright color. The consistent and uniform distribution of each principal element in Al_x_CoCrFeNiTi was confirmed by the EDS mapping. The thicknesses of the rim phase and binder were quite small. The HRTEM images and SAED patterns of the interface between rim and binder are displayed in Figure 4. The interplanar spacing of the selected A area of the rim phase as ahowed in A image was 0.249 nm, being close to that (0.246 nm) between the (111) planes of Ti(C,N). Because the lattice constant and interplanar spacing of a (Ti, W, Mo)(C, N) solid solution increased due to the presence of WC and Mo_2_C solutions, the XRD peak position of the solid solution moved toward the lower angles. According to the SAED analysis, the plane was identified as ($\overset{-}{1}1\overset{-}{1}$) of the (Ti, W, Mo)(C, N) solution. The interplanar spacing of the selected B area of binder as showed in B image was 0.215 nm, approaching that between the (200) planes of FCC phase. Based on the SAED data, it was thus identified as the ($\overset{-}{2}00$) plane of FCC phase, which was consistent with the XRD results of HEAs. In order to contrastively analyze the interface between cobalt binder and rim, the HAADF microstructure images and EDS maps were acquired from TNCo cermet (Figure 5).

Besides, there existed hard phase agglomerates (black), and the distribution of binder phase appeared uneven and thicker, as shown in Figure 5a,b. The magnified Figure 5c exhibited a thick rim structure in TNCo cermet.

### 3.3. Mechanical Properties

The Vickers hardness and fracture toughness of Ti(C,N) cermets with HEA binders and benchmark TNCo cermet are depicted in Figure 6.

As seen in Figure 6a, the Vickers hardness values of Ti(C,N) cermets with Al_x_CoCrFeNiTi binders were higher than that of TNCo cermet. With the increase in Al molar ratio from 0.1 to 1, the average values of Vickers hardness for cermets with HEA binders varied from 1966.5 MPa to 2464.5 MPa, which was because of the formation of hard BCC phase and strong cohesive bonding between Al and other elements [24]. The addition of HEAs inhibited the grain growth, and the appearance of finer grains meant the stronger grain boundary strengthening [25]. At the same time, the improvement in hardness of cermets was owing to the enhancement in solid solution strengthening. By contrast, the hardness of TNCo cermet was 1605 MPa. According to Figure 6b, the fracture toughness values of Ti(C,N) cermets with Al_x_CoCrFeNiTi binders also exceed that of TNCo cermet. Once the Al molar ratio increased from 0.1 to 1, the average value of fracture toughness for cermets with HEA binders rose from 14.9 to 18.2 MPa × m^1/2^, where the maximum value was attributed to that of AlCoCrFeNiTi cermet. Meanwhile, the fracture toughness of TNCo cermet was 13 MPa × m^1/2^. This might be ascribed to the microstructure differences observed in the TEM micrographs: while there was a thin and homogeneous core–rim structure in cermets with HEA binders, thick rims were observed in TNCo cermet. Conforming to some studies, thin rims could have contributed to the low interfacial stress so as to improve fracture toughness [26].

### 3.4. High-Temperature Oxidation Behavior

The oxidation kinetics curves of the five cermets at 1000 °C are plotted in Figure 7. The mass gain of the TNCo cermet was the largest, followed by those of TNAl, TNAl2, and TNAl4. The mass gain of TNAl3 was at the lowest value. The oxidation kinetics curves of TNAl2, TNAl4 and TNAl3 changed according to a quasi-parabolic law. In turn, those of TNAl1 and TNCo tended to a linear law. This might be because the increasing content of aluminum induced severe lattice distortion, leading to grain refinement, which could inhibit the grain growth of hard phase and rim phase. Thus, the most optimal mechanical properties were achieved in TNAl3 and TNAl4 specimens.

The oxidation cross-sectional morphologies and EDS maps of five cermets are shown in Figure 8, including TNAl1, TNAl2, TNAl3, TNAl4, and TNCo. As seen in Figure 8a, titanium and aluminum diffused to the surface layer in TNAl1 cermet and were then distribute in the internal oxidation layer and external oxidation layer.

Given the oxygen distribution (Figure 8a), both of the oxidation layers were discontinuous. As seen in Figure 8b, titanium, chromium and aluminum all diffused toward the surface layer and then accumulated within the external oxidation film in TNAl2 cermet. Although the internal oxidation film was thicker than the external one, both films had a loose structure. In Figure 8c, titanium, chromium and aluminum elements still diffused to the surface layer in TNAl3 cermet. While the external oxidation film was thin and loose, the internal film was a little thicker and possessed the higher compactness, which was confirmed by the oxygen distribution. According to Figure 8d, the diffusion of titanium, chromium and aluminum toward the surface layer in TNAl4 cermet was similar to that in TNAl3 cermet. The external oxidation film was thin but rather compact, whereas the compactness of the internal oxidation film was superior to that in TNAl3, which was proved by the oxygen distribution. In Figure 8e, cobalt diffused to the surface layer of TNCo cermet, remaining in both the internal and external oxidation films. However, the internal film was much thicker than those in HEA-containing cermets, and the inward oxygen diffusion was evident. The key factors impacting on the oxidation resistance of the oxidation fil, include the thickness, continuity, and compactness [27,28]. In particular, the oxidation film with moderate thickness, better continuity and compactness can prevent the inward diffusion of oxygen and the further oxidation of materials thanks to its better oxidation resistance. With the increase in aluminum molar ratio from 0.1 to 1, the internal oxidation film became thinner, and its continuity and compactness increased. Therefore, the oxidation resistance can be enhanced through the increase in the aluminum content. This might be due to the following reasons. First, the grain growth of cermet was restrained by high-entropy alloys. Second, there were the lattice distortion and slow diffusion effect of HEAs [29]. Third, the elements, such as aluminum and chromium, were beneficial to oxidation resistance. Moreover, the oxidation film of TNCo cermet was quite loose. As a result, the oxygen atoms could pass through the pores and combine with internal elements, thereby deteriorating the oxidation resistance of TNCo cermet with cobalt binder relative to cermets with HEA binders.

The XRD patterns after oxidation are shown in Figure 9. While the oxidation products of TNCo were mainly presented by TiO_2_, abundant Ti(O_0.19_,C_0.53_,N_0.32_) intermediate phases were detected in four cermets with HEA binders. In a word, the oxidation of the latter cermets was weakened due to the existence of HEAs. In addition, the complex oxides, including NiMoO_4_, NiWO_4_, FeMoO_4_, Cr_2_O_3_, Fe_3_Ti_3_O_9_ and Ni_3_TiO_7_, formed in cermets with HEA binders, among which the TNAl3 cermet exhibited the lower content of TiO_2_. Although CoWO_4_ existed in the oxidation film of TNCo, the oxidation layers of cermets with HEAs were more protective because of complex oxides, which inhibited the diffusion of oxygen into the cermet and improved the oxidation resistance of the material. Therefore, according to the cross-sectional morphology and phase composition of oxidation layers, the TNAl3 cermet had the superior oxidation resistance compared to other cermets.

## 4. Conclusions

In this study, Al_x_CoCrFeNiTi (x = 0.1, 0.3, 0.6, 1) HEAs powders were prepared via mechanical alloying and used as binders in SPS-produced Ti(C,N) cermets. Meanwhile, the benchmark cermet with cobalt binder was prepared as a reference sample. Based on the findings, the following conclusions can be drawn.

With the increase in aluminum molar ratio, the phase structure of A_lx_CoCrFeNiTi HEAs changed from FCC phase to a mixture of FCC and BCC phases. The cermet grains were refined by HEAs binders, promoting the formation of WC and Mo_2_C solid solutions and making the core–rim structure more homogeneous compared with a cobalt-based cermet.The Vickers hardness and fracture toughness of the cermets increased with the increasing aluminum molar ratio. At the aluminum molar ratio of 1, the Vickers hardness and fracture toughness were 2464.5 MPa and 18.2 MPa × m^1/2^, respectively, while those of cermet with cobalt binder were 1605 MPa and 13 MPa × m^1/2^.After static oxidation at 1000 °C, the mass gain of the cermets with AlxCoCrFeNiTi binders was described by a quasi-parabolic law, where the cermet with Al_0.6_CoCrFeNiTi binder exhibited the lowest mass gain. In turn, the oxidation kinetics curve of the benchmark cermet with cobalt followed a linear law. The oxidation product of Ti(C,N)-based cermets with cobalt showed more TiO_2_, whereas the Ti(C,N)-based cermets with AlxCoCrFeNiTi binders were transformed into complex oxides, such as NiMoO_4_, NiWO_4_, FeMoO_4_, Fe_3_Ti_3_O_9_, and Ni_3_TiO_7_. The oxide layer on the cermet with Al_0.6_CoCrFeNiTi binder appeared to be dense and protective, which inhibited the diffusion of oxygen into the cermet and improved the oxidation resistance of the material.

## Figures and Tables

**Figure 1 materials-16-02894-f001:**
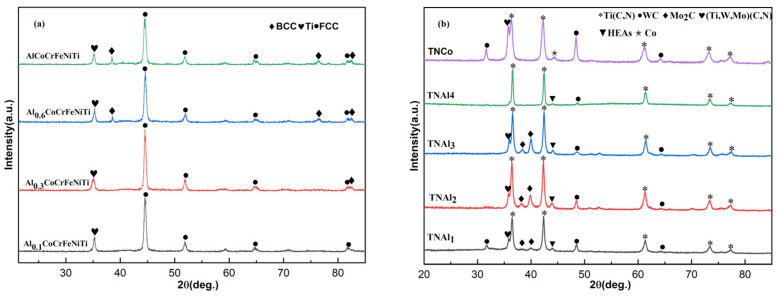
XRD patterns of (**a**) Al_x_CoCrFeNiTi (x = 0.1, 0.3, 0.6, 1) HEA powders and (**b**) sintered Ti(C,N) cermets with different binders (TNAl1, TNAl2, TNAl3, TNAl4, and TNCo).

**Figure 2 materials-16-02894-f002:**
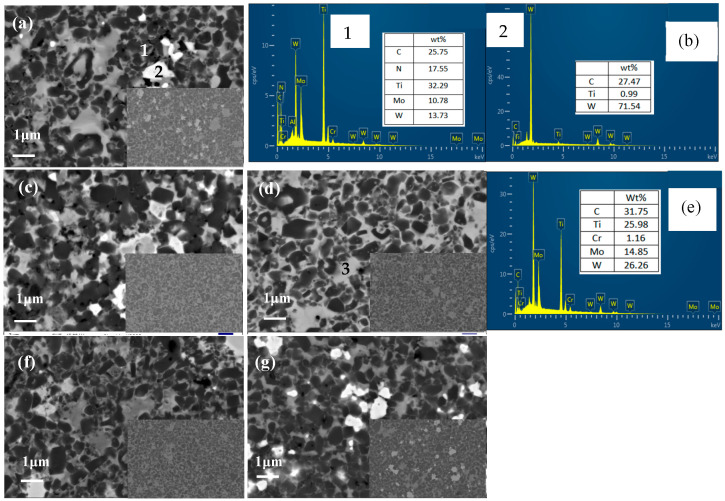
SEM images and EDS analysis of five Ti(C,N) cermets: (**a**) TNAl1, (**b**) EDS analysis of points 1 and 2 in image (**a**), (**c**) TNAl2, (**d**) TNAl3, (**e**) EDS analysis of point 3 in image (**d**), (**f**) TNAl4, (**g**) TNCo.

**Figure 3 materials-16-02894-f003:**
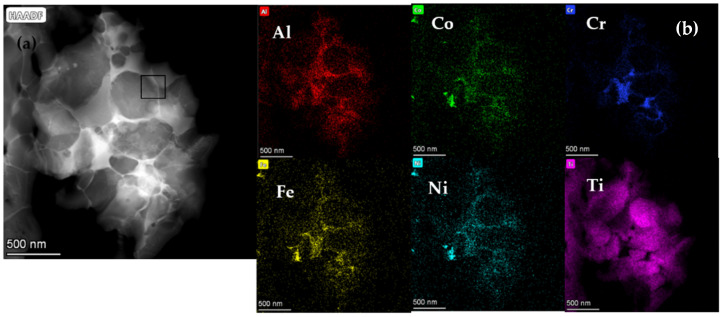
(**a**) TEM/HAADF micrographs of TNAl3 cermet and (**b**) EDS maps of the region shown with a square in image (**a**).

**Figure 4 materials-16-02894-f004:**
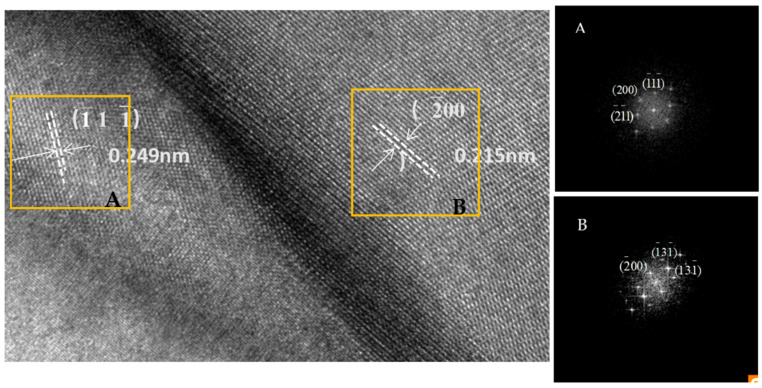
HRTEM micrographs and corresponding SAED patterns of the marked area in Figure 3.

**Figure 5 materials-16-02894-f005:**
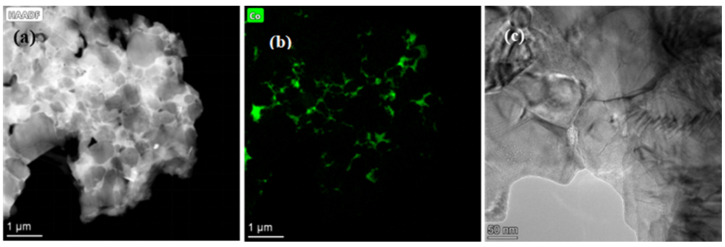
TNCo cermet: (**a**) TEM/HAADF micrograph, (**b**) corresponding EDS map, (**c**) magnified TEM graph.

**Figure 6 materials-16-02894-f006:**
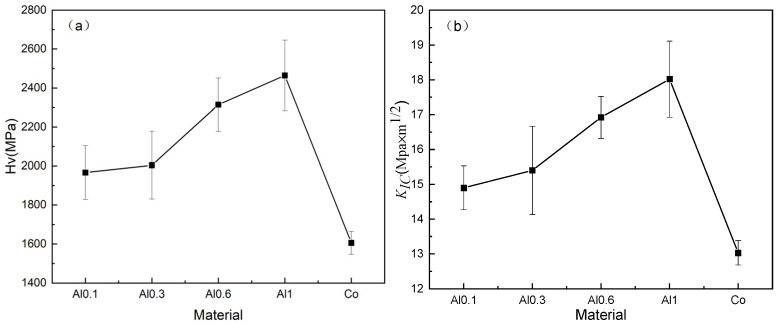
(**a**) Vickers hardness and (**b**) fracture toughness of five cermets.

**Figure 7 materials-16-02894-f007:**
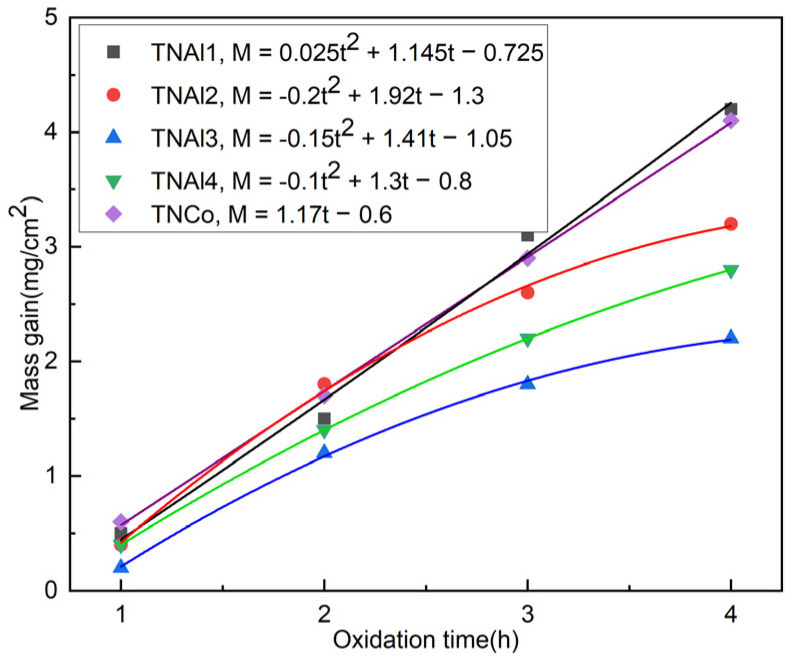
Oxidation kinetics curves of five cermets.

**Figure 8 materials-16-02894-f008:**
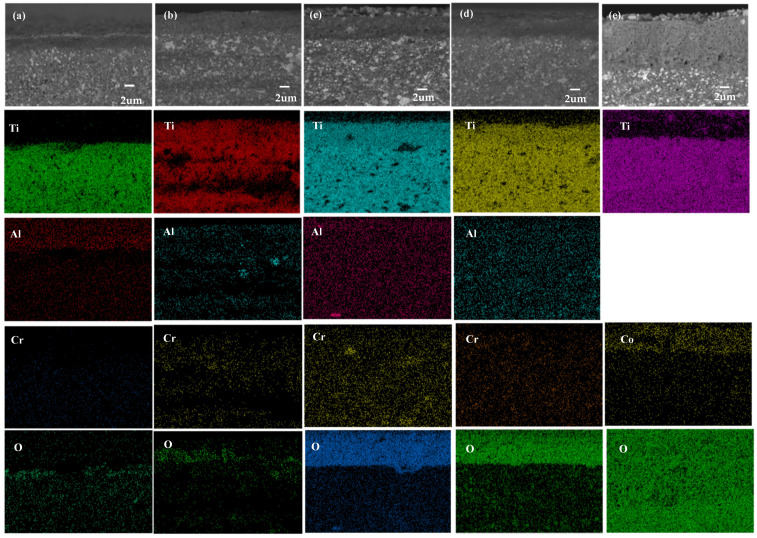
Oxidation cross-sectional morphologies and EDS maps of (**a**) TNAl1, (**b**) TNAl2, (**c**) TNAl3, (**d**) TNAl4, and (**e**) TNCo cermets.

**Figure 9 materials-16-02894-f009:**
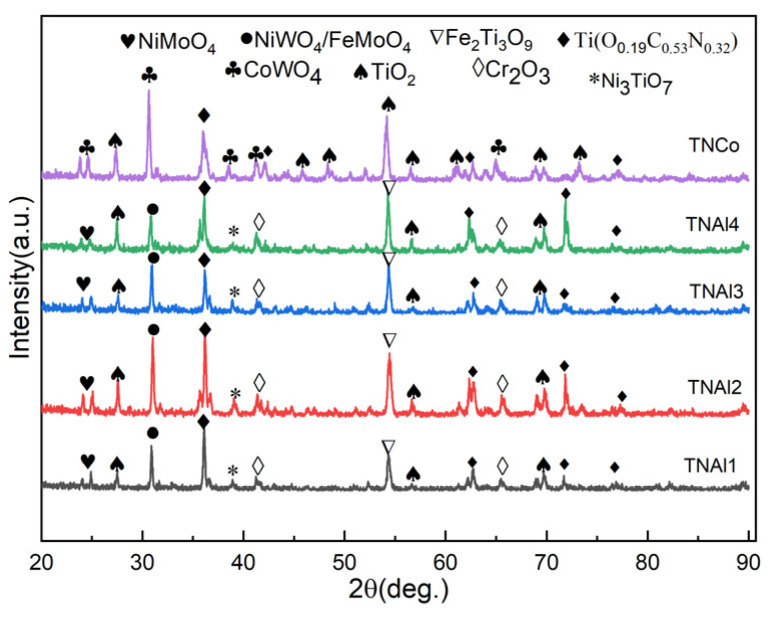
XRD patterns of five cermets after oxidation at 1000 °C for 4 h.

## Data Availability

Not applicable.

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
