# Peer review of "Microstructure and Properties of Ti(C,N)-Based Cermets with Al_x_CoCrFeNiTi Binder"

_materials, 2023, doi:10.3390/ma16072894_

Round 1
Reviewer 1 Report
Some questions and comments to the authors:
1. Why was spark plasma sintering (SPS) chosen as compaction method for the samples?
2. On Fig 2 the bright particles are declared as 'probably' WC (Fig. 2a) and gray areas as 'probably' Mo2C (Fig. 2c). There is no need to speculate as the composition of the particles is quite easy to analyze with SEM-EDS point analysis. The quantitative composition analysis of all phases of the samples should have been quite essential for the current study, but is apparently missing.
3. The EDS mapping images of Fig. 3 are quite difficult to understand. The element labels are too small and unreadable. And isn't it possible to put the colored element images on the background of TEM micrograph for better understanding?
4. Fig 2d refers to 'typical and homogeneous' core-rim structure, but I just cannot see it on the image. SEM image(s) with higher magnification are needed to show the structure of carbide phase grains.
5. The same applies to Fig.3 - there is a lot of guesswork to understand what is actually referred on the image as 'core' and 'rim'. Usually the center of the carbide grain, consisting of more or less pure Ti(C,N) is known as 'core', which is surrounded by 'rim', where Ti(C,N) is combined with more carbides. Does it have the same meaning in the current paper?
6. Isn't there any quantitative methods available to compare the oxidation resistance of the alloys? The EDS mapping images (Fig.7) of the surface layer cross-sections are nice to see and are certainly valuable to understand the oxidation process, but in my opinion the do not substitute direct numerical data.
Author Response
- Why was spark plasma sintering (SPS) chosen as compaction method for the samples?
A: The spark plasma sintering (SPS) is a novel PM method with fast heating and cooling rate and short holding time. The heating is accomplished by electric discharge in voids between particles. SPS can achieve HEAs with high density, fine grains, and good mechanical properties. Due to these discharges, the particle surface is activated
and purifified. A self-heating phenomenon is generated between the particles, as a result of which heat-transfer and mass-transfer can be completed instantaneously.
- On Fig 2 the bright particles are declared as 'probably' WC (Fig. 2a) and gray areas as 'probably' Mo2C (Fig. 2c). There is no need to speculate as the composition of the particles is quite easy to analyze with SEM-EDSpoint analysis. The quantitative composition analysis of all phases of the samples should have been quite essential for the current study, but is apparently missing.
A: The SEM-EDS point analysis has been applied to typical particles and phases to conform the composition.And the results has been updated both in SEM image in Fig.2 and the paragraph.
- The EDS mapping images of Fig. 3 are quite difficult to understand. The element labels are too small and unreadable. And isn't it possible to put the colored element images on the background of TEM micrograph for better understanding?
A: The EDS elements mapping picuture in Fig.3 has been marked more clearly.
- Fig 2d refers to 'typical and homogeneous' core-rim structure, but I just cannot see it on the image. SEM image(s) with higher magnification are needed to show the structure of carbide phase grains.(高倍)
A : The 20K magnification image has replaced the 10K magnification, and the latter one is placed as inner set.
- The same applies to Fig.3 - there is a lot of guesswork to understand what is actually referred on the image as 'core' and 'rim'. Usually the center of the carbide grain, consisting of more or less pure Ti(C,N) is known as 'core', which is surrounded by 'rim', where Ti(C,N) is combined with more carbides. Does it have the same meaning in the current paper?
A: Actually, it has the same meaning as in the current paper. The undissolved Ti(C,N) is core, and the surrounded carbonitride solid solution is rim. The morphology had been identified in SEM, and the selected area and particle in TEM image also exhibit core-rim structure. As SEM-EDS in Fig.2 helps to confirm the phase composition of core-rim structure.
- Isn't there any quantitative methods available to compare the oxidation resistance of the alloys? The EDS mapping images (Fig.7) of the surface layer cross-sections are nice to see and are certainly valuable to understand the oxidation process, but in my opinion the do not substitute direct numerical data.
A:Yes, the oxidation kinetics curves has been drawn and the numerical data for oxidation process has been illustrated.
Reviewer 2 Report
The proposed work focuses on the Microstructure and Properties of Ti(C,N) based Cermets with AlxCoCrFeNiTi Binder. It is of potential interest to Materials journal readers.
Despite the importance of the subject addressed, this work needs many improvements to be ready for the publication in the Materials journal.
Specific points of improvement :
- Abstract is too short. This section must more developed.
- Literature review section must be improved by more previous researches.
- The objective of this research must be more developed by a comparison with previous researches results.
- All test standards must be indicated in Materials and Methods section
- Results and discussions need an in-depth discussion.
- Quality of figures must be improved.
Author Response
Specific points of improvement :
- Abstract is too short. This section must more developed.
Yes, the section has been developed.
- Literature review section must be improved by more previous researches.
For introduction part ,the number reference of has not been put up more.The relevant maybe are considered more important than the references amounts.
- Theobjective of this research must be more developed by a comparison with previous researches results.
Yes, the comparision has been made to highlight the objective of this research.
- All test standards must be indicated in Materials and Methods section
Yes ,the test standard of mechanical properties has been added.
5.Results and discussions need an in-depth discussion.
Yes, more discussion has been developed as in revision edition.
6.Quality of figures must be improved.
Yes, the figures have been treated again and quality has been improved, as showed in revision edition.
Reviewer 3 Report
The work is devoted to the study of the applicability of AlxCoCrFeNiTi powders obtained by the mechanochemical alloy method as a binder component for the production of cermets based on titanium nitride and carbide. In general, the presented direction is very interesting and promising in modern materials science, since the studied cermets have a great potential for use in various industries, including the energy sector. The authors have done a lot of work to study the properties of the obtained structures, including the determination of oxidation resistance, as well as the mechanical properties of ceramics. In general, the claimed article corresponds to the subject of the claimed journal, and also contains new, previously unidentified facts and data about these materials. The article can be accepted for publication after the authors answer a number of questions from the reviewer and provide the corrected version of the article to the editors.
1. The presented data on the dimensions of interplanar distances in Figure 4 are not explicitly expressed, the authors are invited to provide a more detailed image of this area and indicate these dimensions, as well as their correspondence to X-ray diffraction data.
2. The presented data of transmission electron microscopy clearly shows the presence of various types of grains that differ not only in shape but also in structure, as well as elemental compositions, the authors should pay attention to this.
3. The results of X-ray phase analysis require additional data in the form of a phase diagram, as well as an assessment of the contributions of various phases and structural parameters, and the authors should also pay attention to the effect of component variation on the degree of structural ordering.
4. For the expression used to determine the crack resistance, the authors should provide a decoding of all used letter designations taken from [22].
5. The results of increasing the resistance to oxidation of the studied samples, according to the statement of the authors, showed that cermets not containing cobalt have the greatest resistance. The authors attribute this to the presence of an oxide layer, in this regard, a more detailed description should be given of what kind of oxide layer can be present and what is its role in the oxidation process.
6. Hardness and fracture toughness results reflect a significant increase with composition change with the addition of aluminum and a sharp deterioration with the addition of cobalt. In this regard, more explanations for this phenomenon should be given, as well as the reasons for this decline.
7. For technical comments: the authors need to check the text for typos and grammatical errors found in the text.
Author Response
- The presented data on the dimensions of interplanar distances in Figure 4 arenot explicitly expressed, the authors are invited to provide a more detailed image of this area and indicate these dimensions, as well as their correspondence to X-ray diffraction data.
The enlarged detailed TEM image of this area is presented.
- The presented data of transmission electron microscopy clearly shows the presence of various types of grains that differ not only in shape but also in structure, as well as elemental compositions, the authors should pay attention to this.(OK)
The EDS elements mapping picuture in Fig.3 has been marked more clearly。
- The results of X-ray phase analysis require additional data in the form of a phase diagram, as well as an assessment of the contributions of various phases and structural parameters, and the authors should also pay attention to the effect ofcomponent variation on the degree of structural ordering.
I couldn’t find the phase diagram of solid solution reaction of Ti(C,N) , WC and Mo2C. But the BCC-to-FCC peak area ratios and (Ti, W, Mo)(C, N)-to-Ti(C, N) peak area ratios were calculated to assess the contributions of various phases. It’s proved that (Ti, W, Mo)(C, N)-to-Ti(C, N) peak area ratios inceased with the increasing of Al content in HEAs. And also,the grain size of Ti(C,N) in five cermets measured from the SEM micrographs using the linear interception method were given out. The gain size decreased with the the increasing of Al content.
- For the expression used to determine the crack resistance, the authors should provide a decoding of all used letter designations taken from [22].
It’s finished.
- The results of increasing the resistance to oxidation of the studied samples, according to the statement of the authors, showed that cermets not containing cobalt have the greatest resistance. The authors attribute this to the presence of an oxide layer, in this regard, a more detailed description should be given of what kind of oxide layer can be present and what is its role in the oxidation process.
- The phase composition of oxidation surface has been shown in XRD image and manuscript.
- Hardness and fracture toughness results reflect a significant increase with composition change with the addition of aluminum and a sharp deterioration with the addition of cobalt. In this regard, more explanations for this phenomenon should be given, as well as the reasons for this decline.
The Ti(C,N) cermet with AlxCoCrFeNiTi HEAs binder and the cermet with cobalt binder are benchmark tests samples. The former ones showed better mechanical properties than the later one, and it’s not increasing first and then decline. The explanation was also further imparoved with more reference.
7.For technical comments: the authors need to check the text for typos and grammatical errors found in the text.
The native English-speaking editing has been finished.
Round 2
Reviewer 1 Report
No comment
Reviewer 2 Report
The proposed work focuses on the Microstructure and Properties of Ti(C,N) based Cermets with AlxCoCrFeNiTi Binder. It is of potential interest to Materials journal readers.
I think that the revised version of the submitted paper is well improved by considering the reviewers and editor recommendations and remarks. Indeed, I think that this paper is accepted in this form.
Reviewer 3 Report
The authors answered all the questions, the article can be accepted for publication.